# Impact of Lifting of Two Types of Barrels on Postural Control, Trunk Muscle Recruitment, and Kinematic Measures in Manual Workers

**DOI:** 10.3390/ijerph16122183

**Published:** 2019-06-20

**Authors:** Amanda M.S. Cavaguchi, Márcio R. Oliveira, Christiane G. Macedo, Pablo E.A. de Souza, Andreo F. Aguiar, Mathieu Dallaire, Suzy Ngomo, Rubens A. da Silva

**Affiliations:** 1Master and Doctoral Programs in Rehabilitation Sciences, UEL/UNOPAR, Londrina 86083-070, PR, Brazil; amanda.fisio@outlook.com (A.M.S.C.); marxroge@hotmail.com (M.R.O.); chmacedouel@yahoo.com.br (C.G.M.); pablo.eduardo.a.souza1@gmail.com (P.E.A.d.S.); afaguiarunesp@gmail.com (A.F.A.); 2Physical Therapy, Universidade Estadual de Londrina, Londrina 86057-970, PR, Brazil; 3Département des Sciences de la Santé, Programme de physiothérapie de l’Université McGill offert en extension à l’Université du Québec à Chicoutimi (UQAC), Centre intersectoriel en santé durable, Laboratoire de recherche BioNR -UQAC, Saguenay, Québec, G7H 2B1, Canada; mathieu.dallaire2@uqac.ca (M.D.); Suzy_Ngomo@uqac.ca (S.N.)

**Keywords:** kinesiology, trunk muscles, posture, biomechanics

## Abstract

The aim of this study was to evaluate the impact of 2 types of beer barrels on postural control, trunk activation, and kinematic measures in adult workers. Twelve (12) males randomly performed 4 tasks on a force platform for 20 s: (1) hold an empty recyclable barrel, (2) hold a full recyclable barrel (30 L), (3) hold an empty steel barrel, and (4) hold a full steel barrel (30 L). Trunk muscular activation, force platform and kinematic measures at the trunk, hip, and knee joints were computed. The full steel barrel produced greater postural oscillation than other conditions. Higher trunk activity was also reported during the full steel barrel task. Significant kinematic changes only in the trunk were observed between the empty steel barrel and the full recyclable barrel tasks. In conclusion, the full steel barrel produced a negative impact on postural control, increasing trunk activity and changing trunk flexion angle in adult workers.

## 1. Introduction

Some evidence reports the negative impact of physical tasks on health in the genesis of occupational injuries. Musculoskeletal disorders (MSD) account for a significant proportion of occupational injuries. The main physical risk factors for work-related MSD are excessive physical overload or repetitive movements associated to compressive forces on spine during lifting, trunk flexion, and awkward postures [1,2]. In addition, muscle weakness and fatigue of lumbar [3,4] and postural instability through the uncoordinated action of the trunk muscles are also considered primary risk factors associated with MSD such as low back pain (LBP) and disabilities over time [5,6,7]. 

The association between trunk function and work overload is of interest in determining preventative actions to avoid MSD. It is known that people who have suffered from low back pain have a propensity to recurrence, therefore ergonomic or functional conditions in the workplace and at home still need to be studied to continue to improve on work operations. For example, a recent study from our laboratory [8] showed the negative effect of holding an external load (10% of body mass) on postural stability in both young and older adults with and without chronic low back pain. These results reinforce the idea that holding external loads can impair standing balance as well as increase the risk factors for injuries and degeneration of the vertebral column over time [9,10,11].

Furthermore, Jones and collaborators [12] reported important risks of spinal injuries associated with work and overload demands in the employees of a brewery/bar. The authors showed that elevated compressive (8660.4 N) and shear (752 N) forces at the lumbosacral joint are required to lift the keg, and that only 3.6% of the female worker population would have the necessary trunk strength to accomplish this task, and that 100% of the population would not have the shoulder or elbow strength required. In addition, lumbosacral joint compression values exceed the maximum permissible limit of 6700N recommended by Waters et al. [13], suggesting that lifting resulting in this degree of compression would be hazardous for most people (75% of men and 99% of women). Thus, one of the authors’ main recommendations to minimize this condition was to redefine the keg design (format, weight etc.). All this knowledge provides a better understanding of the biomechanical mechanisms involved in low back pain at work [14,15] and secondly, should help in the implementation of ergonomic improvement processes

A labor product recently manufactured in Brazil suggested a keg called a Pet barrel, which is a lighter, stronger, safer, and more economical barrel (24 November 2017 listed on its website www.beerkeg.com.br). This raises a research question: could this lighter barrel be an alternative to minimize the burden on professional tasks related to trunk efforts and musculoskeletal disorders in this work context? In fact, a more comprehensive study is warranted to assess the impact of this type of barrel on trunk neuromuscular and postural responses, and consequently to be an alternative to prevent MSD in workers. Therefore, the purpose of this study was to compare the effects of two types of barrels, recyclable versus steel, on neuromuscular, postural and kinematic trunk responses in workers without LBP. We hypothesized that the negative impact of the steel barrel would be greater than the recyclable barrel for both empty and full conditions. 

## 2. Method

### 2.1. Participants

The sample for this study was of 12 male volunteers who were recruited for convenience from local communities. In Brazil, lifting of loads at work is generally observed in jobs occupied by men than women, which justify only males in the present study for this demonstraiton. The characteristics of participants were in mean: age =27 (Standard Deviation: SD = 4), height =1.69 m (SD = 14), mass = 69 kg (SD = 14), number of working days per week =6 (SD = 2), number of working hours per week =35 (SD = 3), number of working months in the same job associated to lifting loads =77 (SD = 40). As no study was found comparing barrel types, calculation of the sample was based on a similar study made by Shigaki et al. [8], which they assessed the effects of a trunk load (10% of body weight) on standing posture stability measures in young and old adults with and without chronic low back pain. Data of the mean of the group of healthy young people was used with the most sensitive and reliable variable to determine the present sample, namely velocity of oscillation of pressure center. The estimated values were from differences of the mean between no load and with load (0.39 ± 0.30 cm/s), with power of the test at 95% and statistical alpha at 5% (*p* < 0.05) during a bilateral test of paired samples. With these calculated values, a total of 10 individuals should have been recruited. However, when considering a margin loss of 15%, the study enrolled 12 subjects to demonstrate the effect of barrels with and without load (30 L).

The inclusion criteria for the participants were: good general health, active in the labor market and experienced with such labor demands, and that they be males. The exclusion criteria were neurological, orthopedic, cardiac and/or metabolic diseases; vestibulopathies and labyrinthine crises; mental disorders, attention and speech disorders; and locomotor surgery because all these factors could influence our measures.

Each participant was informed about experimental procedures before giving written consent. The research project was approved by the local Research Ethics Committee (CER: 1.966.838)

### 2.2. Procedures

All experimental procedures were conducted by trained physiotherapists enrolled in the study (A.M.S.C. and P.E.A.d.S.). Basic evaluation steps were as follows: (1) anamnesis and anthropometric measures, (2) familiarization with the experimental conditions and weights of the barrels, (3) electromyography (EMG) surface electrodes and markers (Kinematics) placement, (4) Maximal Voluntary Contraction (MVC) task for EMG normalization procedures, and (5) experimental protocol performance with both types of barrel (Figure 1). Data collection lasted for approximately 2 h maximum.

### 2.3. Instrumentation

#### 2.3.1. Electromyography

The electromyographic signal capture system was the Bagnoli-8, Delsys system, Inc. (Wellesley, MA, USA). The EMG signal was captured with 8 pre-amplified active electrodes (gain: 1000) and filtered through a bandpass between 25 and 450 Hz with a sampling frequency of 2000 Hz.

Electrodes were positioned on trunk muscles (bilaterally); especially in the ilio-costal at the L3 vertebral level (EMG-L3), multifidus at the L5 vertebral level (EMG-L5), and in the abdominal muscles (external oblique: EMG-EO and rectus abdominal: EMG-RA). The maximum voluntary contraction (MVC) reference test was used for EMG data normalization and to determine the trunk activation recruitment pattern under four experimental conditions. The electrodes were positioned in relation to the orientation of the muscle fibers of the respective trunk muscles, as performed by Larivière, Gagnon and Loisel [16]. The reference electrode was positioned in the vertebral spinal process of T8.

All EMG signals were processed and treated with MATLAB program routines (Version 11.0; The MathWorks Inc., Natick, MA, USA) to extract the electromyographic indicators of muscle activation, such as the EMG signal amplitude in the Root Mean Square: RMS. To calculate the RMS parameter, the EMG signal was processed by 250 ms overlapping windows to determine the temporal information of the event, and to define then, the mean RMS amplitude (RMS_TASK_) duringto the task time for each muscle group (paravertebral and abdominal groups) [16].

The same RMS computation procedure was employed for the MVC protocol, but with 100 ms temporal windowing due to the maximal 5-s duration. From 2 MVC trials by each muscle group, the highest value of RMS (peak) was considered as representative of the reference value (RMS_MAX_) for normalization. This procedure was applied to each muscle group in each experimental condition (please see: experimental protocol section). The final equation to determine trunk activation pattern during tasks on the force platform was:% RMS − EMG = [(RMS_TASK_ / RMS_MAX_) × 100%](1)

This method is valid to compare groups and muscles and has been an important guideline in the methodological procedure with EMG based on literature of issue [17]. 

#### 2.3.2. Force platform (Postural Stability)

The system used to measure postural stability in the different experimental tasks was the BIOMEC400 force platform (EMG System do Brasil, Ltda.). The vertical ground reaction force data from the force platform (Z forces) were sampled at 100 Hz. All force signals were filtered with a 35 Hz low-pass second-order Butterworth filter [18]. The signals from the force platform sensors were converted into Centre of Pressure (COP) data using computerized stabilography using MATLAB routines (The Mathworks, Natick, MA). The following balance parameters were calculated: Total displacement of the foot pressure center (maximum COP displacement in cm), area ellipse of COP to 95% confidence interval (A-COP cm^2^), sway velocity (VEL) of COP in the anteroposterior direction (A/P), and mediolateral (M/L) movement. The reliability (ICC > 0.80) of COP parameters is acceptable to assess postural stability or postural control [18], which support to use in the present study.

To control the effects on the characteristic variations of each barrel, all force platform data based in COP domain (ellipse area: A-COP and oscillation velocity: A/P and M/L VEL) were normalized by total displacement of COP (in cm) and adjusted according to the characteristics of each barrel (measured in cm of diameter and height). This was applied for the first time for these variables so that to avoid a determinant bias on the protocol that is related to barrel design (their measure and dimension). To determine an optimal reference (denominator) for this normalization procedure, the total displacement (in cm) on force platform was added to diameter and height values of each barrel in cm for a single factor of correction. A final equation of COP normalized was:% COP = [COP_cm_/factor_cm_) × 100%](2)

Note: factor (total displacement + diameter-barrel + height-barrel) was applied for each COP variable (Area and velocity).

### 2.4. Kinematics (Photogrammetry)

Participants received markers on specific joints recommended by the postural assessment program (SAPO) to determine the angulation changes of each joint. For each experimental condition in which the participant held the barrel, the position was photographed. For the analysis we used an Apple^®^ brand camera, Model iPhone 6. The camera was fixed to a tripod located on the right side of the individual at a distance of 2 m with a height of 1 m and 50 cm from the ground. The angles analyzed were: Angle of the knee (union between the major trochanter points, knee art line and lateral malleolus, hip angle (formed between L5, trochanter and knee line), and trunk oscillation angle (formed by the union of C7, trochanter and malleolus). All measurements were performed with reference to the 0°. For the knee angles, positive values refer to knee hyperextension, while negative values represent flexion. For the hip angles, the values presented closer to zero represent the hip flexion movement. Positive values for trunk angles refer to flexion, while negative values represent trunk extension. Figure 2 illustrates the marking points from the analyzed joints (knee, hip, and trunk angle).

### 2.5. Pedometer (Physical Activity Level)

In agreement with Tudor-Locke and Bassett [19], a pedometer (Yamax Digi-Walker^®^ Pedometer—model SW 700) was used for one week to determine the physical activity level of participants; the necessary information was recorded on a daily chart containing all the information about the device as well as user and contact information for the researcher in charge. According to Tudor-Locke and Bassett [19], based on available evidence, the following preliminary indices were used in order to classify the level of physical activity determined by the pedometer in healthy of participants in function of the number of steps: <5000 = sedentary, 5000–7499 = poorly active, 7500–9999 = moderately active; 10000–12500 = active, >12500 = very active.

### 2.6. Experimental Protocol

The experimental protocol was performed in a biomechanics laboratory in a quiet environment with the temperature set at 23 °C. First, each participant performed MVC efforts for EMG normalization procedures [16]. Each participant performed two MVCs for 5 s, with an interval of 1 min in between. The interval between muscle groups (abdominal and paravertebral) was also 1 min. The postures adopted by the participants during the tests were dorsal decubitus with flexion of the legs for the abdominal muscle test and in prone position with legs extended for the paravertebral muscle test. The participants were stabilized well to better isolate the muscle group evaluated.

After MVC, the participant was positioned on the force platform to carry out the 4 experimental conditions. These experimental conditions were presented randomly:(1)Support empty steel barrel (Figure 3A);(2)Support full steel barrel (Figure 3B);(3)Support empty recyclable barrel (Figure 3C);(4)Support full recyclable barrel (Figure 3D).

Participants were also instructed to stand with their feet fixed on the power platform marking looking at the point affixed to the wall and slightly flex their knees when performing the tasks. To standardize the type of footprint, the participants used both hands to lift the barrels by a single handle. In fact, our preliminary pilot study (*n* = 10) has shown no significant differences (t-test for paired samples, *p* > 0.05) for the steel barrel when comparing the traditional footprint (arms and open hands) with both hands closed. 

During the experiment, the barrels were positioned on the ground in front of the force platform according to the order of the draw. At the commander’s request, the participant began the task by taking the barrel from the ground, held it for 10 s maximum, and returned it to the ground. Each task was performed only once in order to avoid muscle fatigue. Five (5) minutes of rest was applied between each task. At the end of test, a BORG scale from 0 to 10 was applied to measure effort perception across each task. During testing, an EMG record was made simultaneously with the force platform measurement and synchronized through a trigger.

### 2.7. Statistical Analysis

Descriptive analysis was performed with measures of central tendency, mean, and standard deviation. The parametric distribution of the data was verified by the Shapiro–Wilk test. Once data normality was confirmed, an analysis of variance (ANOVA) of a factor (experimental conditions) was used to determine the differences between the 4 experimental conditions [(1) support empty steel barrel, (2) support full steel barrel (3) support empty recyclable barrel, (4) support full recyclable barrel] for the parameters of postural stability, EMG parameters in each muscle group and kinematics measures. When necessary, a Tukey Post hoc test was used to locate the differences between conditions. The effect size was also calculated between the experimental conditions. Effect size (ES) as a statistical concept allows the establishment of a real difference between groups, where a value <0.19 is considered insignificant, between 0.20–0.49 a small effect, between 0.50–0.79 average, between 0.80–1.29 a large effect, and >1.30 a very large effect [20]. The same procedure was used to analyze the kinematics. The statistical program SPSS (version 20.0 for Windows) was used to perform all statistical analyses. The significance was 5% (*p* < 0.05).

## 3. Results 

All participants (*n* = 12) were included in the final analysis. They were classified as very actives from pedometer measures while all presented high levels of physical activity (number of steps in mean =12.965; SD = 4.484).

The results between the experimental conditions for the COP parameters normalized by the total displacement including the characteristics of each barrel are presented in Table 1. Significant differences were found among the 4 conditions for the 3 main variables analyzed (*p* < 0.01). In general, Tukey’s test showed significant differences (*p* < 0.01) mainly between the full steel barrel in relation to the other experimental conditions. The magnitude of effect (Effect Size: ES) for the differences between full streel barrel versus full recyclable barrel was very strong (*d* = 1.06) for the variable A-COP_normalized_, moderate (*d* = 0.6) for VEL/AP_normalized_, and strill strong (*d* = 1.0) for the variable VEL M/L_normalized_. The full steel barrel produced greater postural instability than the recyclable barrel both in empty and full conditions.

The results between the experimental conditions for the EMG parameters normalized by the maximum activation of each muscle are shown in Table 2. Significant differences were found among the 4 experimental conditions for the 4 muscles analyzed (*p* < 0.01). In general, Tukey’s test found significant differences (*p* < 0.01), mainly between the full steel barrel in relation to the other experimental conditions for all muscles, being characterized by greater muscular activation. Effect size regarding the differences between full steel barrel versus empty steel barrel and empty recyclable barrel was on average *d* = 1.41 for the L5 muscle (very strong magnitude of effect). Although no significant difference was found between full steel barrel versus full recyclable barrel, the clinical percentage of the difference between the 2 conditions was on average 10% for all muscles and always higher for the full steel barrel. The perception of the effort obtained during the test was in mean 3/10 for the lumbar region and 7/10 for the upper limbs across the participants for 2 main conditions with full barrels (but without significant differences: results not reported here *p* > 0.05).

The results between the experimental conditions for the 2D kinematic parameters (two dimensions) from the photogrammetry measurements (knee, hip, and trunk angles in the sagittal plane) are shown in Table 3. Significant differences were found between the experimental conditions only for the trunk angle measurements (*p* < 0.01). In general, Tukey’s test found significant differences (*p* < 0.01) only between the empty steel barrel (2°) and full recyclable barrel (−11°). The magnitude of the effect size for the differences between steel and recyclable was an average of strong (*d* = 2.6). All kinematic measures were observed with respect to 0° as reference, so that the effort with the empty steel barrel produced greater vertical trunk flexion than the full recyclable barrel.

## 4. Discussion 

The purpose of this study was to evaluate the impact of 2 types of barrels (recyclable and steel) on trunk neuromuscular pattern responses, postural control, and trunk kinematics in working adults. Our hypothesis was confirmed; the negative impact of the steel barrel was more significant than that of the recyclable barrel in both empty and full conditions. The support steel barrel presented increased postural instability, higher recruitment of the lumbar muscles, and increased postural angle of the trunk in flexion when compared to the task of supporting the recyclable barrel. 

To the best of the authors’ knowledge, this is the first comparison of 2 types of barrels based on biomechanical trunk measures. In the present study, the steel barrel, even in the empty condition, represented 18% of the average body weight, exceeding the 10% recommended by Shigaki [8] of postural stability measures. Under the same conditions, the recyclable barrel represented only 1%. Under the full experimental conditions, the 2 barrels exceeded the values; the steel barrel presented a load of 60% versus 43% for the recyclable barrel. Although the subjective perception of effort in the present study was low for the lumbar region (3/10) as compared to the upper-limbs (7/10) (results not presented here), the impact of this load for the full barrels was similar for effort perception. However, the negative impact measured for the steel barrel compared to the recyclable barrel was non-negligible from a work prevention perspective. In fact, there was a cause–effect relationship between excessive back physical exertion and occurrences of acute and chronic symptoms of LBP [21,22]. Therefore, an excessive external load over the trunk area and prolonged static postures can overload the passive structures of the spine, causing discomfort, physical stress, and inflammatory responses, resulting in LBP [23,24,25]. 

Our data add to the existing literature, which showed postural instability and greater muscle activity of trunk extensors when an external load is supported [8,12,26,27,28]. For example, Hendershot et al. [26] found increased postural instability with a 3-kg external load in healthy young adults during standing tasks of trunk flexion on a force platform. In addition, Grondin et al. [29] reported that muscle activity of the trunk extensors was greater when sudden hand loading was applied, resulting in spine fatigue and imbalance. The sudden unloading also resulted in increased COP displacements in backward direction and decreased postural stability [30]. Our results combined with the studies mentioned above reveal that excessive external loads such as a steel barrel may impair postural control, even for a short period of time (10 s), reinforcing the idea that the development of alternative strategies to mitigate the impact of overload on the trunk muscles is essential to prevent the onset of work-related muscle disorders. 

Several factors are associated with postural instability of the trunk extensors or balance when an external load is supported. For example, when loads are sustained for short or prolonged periods or repeatedly, there are brief uncontrolled intervertebral movements because of back muscle fatigue, which in turn causes postural instability [5]. Holding an external load for 10 s, as in the present study, may contribute to increased trunk flexion, lumbar activity, and consequent increase in fatigue of the trunk muscles that may affect postural control and balance [31,32]. In addition, a decrease in proprioceptive impulses from fatigued trunk muscles results in wider movements of the lumbar spine and changes in muscle activation, resulting in greater postural sway and delayed muscle response time [33]. 

Finally, some limits of this study should be addressed. The results were generalized only for males and healthy active adults and not individuals with low back pain. No dynamic kinematics measurements were used, only photogrammetry method was applied, which limited the interpretation of results on dynamic angles measures. Another factor to consider is that the investigated situation refers only to an experimental investigation in a biomechanics laboratory. It is suggested to generalize this evaluation in the practical environment of workers.

More studies are necessary to evaluate the impact of these barrels on different populations and in different work contexts.

## 5. Conclusions

In conclusion, our findings showed that the use of the pet barrel may attenuate neuromuscular overload on trunk muscles in both empty and full conditions compared to conventional steel barrels, indicating that recyclable pet barrels may be an adequate strategy to promote good health and safety for workers in the beer industry. The present study has important economic and practical implications for the assessment of LBP and changes in the commercialization of beer barrels in the industry to reduce the risk of injuries and worker withdrawal.

## Figures and Tables

**Figure 1 ijerph-16-02183-f001:**
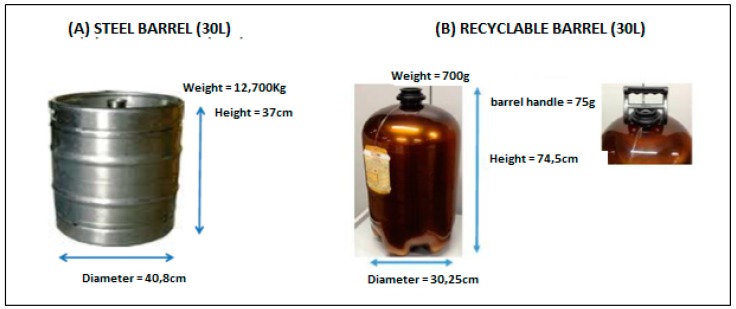
Illustration of the two types of barrels tested. Note: (**A**) steel; and (**B**) recyclable–innovative model. The two with capacities for 30 L.

**Figure 2 ijerph-16-02183-f002:**
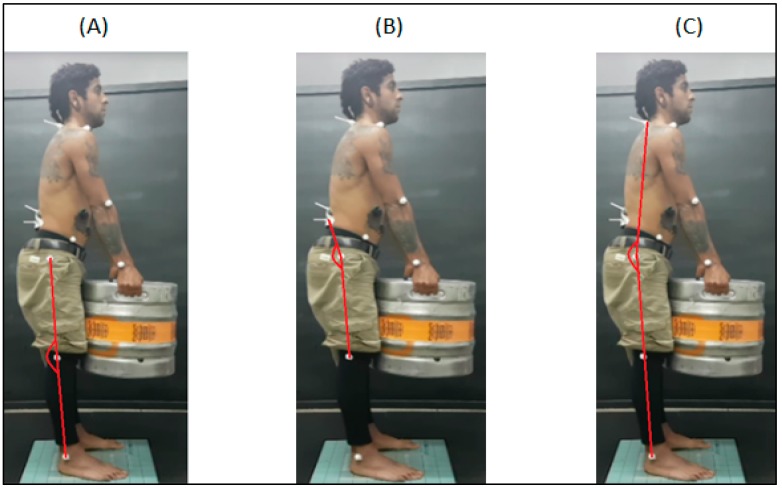
Illustration of the angles analyzed in photogrammetry. Note: (**A**) knee angle; (**B**) hip angle; and (**C**) trunk angle of oscillation.

**Figure 3 ijerph-16-02183-f003:**
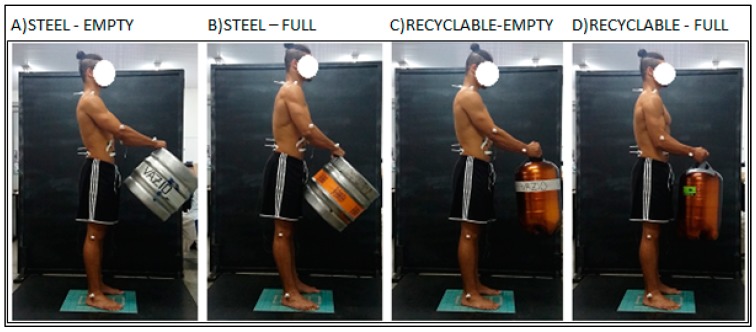
Illustration of the four variables evaluated. Note: (**A**) steel barrel—empty; (**B**) steel barrel—full; (**C**) recyclable barrel- empty; (**D**) recyclable barrel-full.

**Table 1 ijerph-16-02183-t001:** Comparisons of postural control measures through the four experimental conditions.

Variables	Experimental Conditions				One-Way ANOVA	ES
	1	2	3	4	*p-*values (F-value)	*(d)*
A-COP_Normalized_	21.9 (6.6)	42.9 (16.8)	11.0 (3.3)	24.5 (15.0)	<0.01 (21.9)	1.06
					* 2 > 1, 3 and 4 3 < 1,2 and 4 4 > 3 and < 2	
VEL A/P_Normalized_	2.1 (0.5)	4.3 (2.0)	1.5 (0.2)	1.6 (0.5)	**0.03 (3.05)**	**0.6**
					^*^ 2 > 3 and 4	
VEL A/P_Normalized_	1.2 (0.1)	3.2 (2.5)	0.9 (0.1)	1.2 (0.1)	0.169 (1.80)	**1.0**

Note: Mean values and standard deviation in parenthesis. Bold character = significant differences on conditions factor from ANOVA (*p* < 0.05). * Post Hoc analysis (Tukey test) to localize the differences between conditions. Experimental conditions: 1 = steel barrel–empty; 2 = steel barrel– full; 3 = recyclable barrel– empty; 4 = recyclable barrel–full.

**Table 2 ijerph-16-02183-t002:** Comparisons of the trunk muscles EMG measures through the four experimental conditions.

Variables	Experimental Conditions				One-Way ANOVA	ES
	1	2	3	4	*p*-Values (F-Value)	*(d)*
RMS-EMG –L5 (%)	23 (10)	46 (18)	18 (10)	31 (10)	**<0.01 (9.726)**	**1.41**
					^*^ 2 > 1, 3 and 4	
RMS-EMG-L3 (%)	14 (7)	24 (11)	10 (6)	18 (8)	**< 0.01 (5.678)**	**0.80**
					^*^ 2 > 1 and 3	
RMS-EMG-RA (%)	18 (9)	30 (14)	14 (7)	24 (12)	**< 0.01 (4.554)** ^*^ 2 > 1 and 3	**0.83**
RMS-EMG-OE (%)	23 (10)	40 (13)	18 (8)	30 (13)	**< 0.01 (7.680)** ^*^ 2 > 1 and 3	**0.92**

Note: Mean values and standard deviation in parenthesis. Bold character = significant differences on conditions factor from ANOVA (*p* < 0.05). *Post Hoc analysis (Tukey test) to localize the differences between conditions. Experimental conditions: 1 = steel barrel–empty; 2 = steel barrel–full; 3 = recyclable barrel–empty; 4 = recyclable barrel–full. RMS: Root Mean Square, EMG: electromyography, ES: Effect Size.

**Table 3 ijerph-16-02183-t003:** Comparisons between the four experimental conditions in kinematic measures (absolute angles) of the knee, hip and trunk.

Variables	Experimental Conditions				One-Way ANOVA	ES
	1	2	3	4	*P* values (F value)	*(d)*
knee angle (°)	4 (5)	2 (4)	2 (3)	−1 (4)	0.254 (1.422)	**0.05**
hip angle (°)	–25 (10)	−32 (12)	−29 (15)	−35 (11)	0.351 (1.131)	**0.01**
trunk angle (°)	2 (6)	−7 (7)	−1 (12)	−11 (12)	**<0.01 (4.350)** ^*^ 1 ≠ 4	**2.6**

Note: Mean values and standard deviation in parenthesis. Bold character = significant differences on conditions factor from ANOVA (*p* < 0.05). *Post Hoc analysis (Tukey test) to localize the differences between conditions. Experimental conditions: 1 = steel barrel–empty; 2 = steel barrel– full; 3 = recyclable barrel– empty; 4 = recyclable barrel–full.

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
