# Peer review of "Impact of Lifting of Two Types of Barrels on Postural Control, Trunk Muscle Recruitment, and Kinematic Measures in Manual Workers"

_ijerph, 2019, doi:10.3390/ijerph16122183_

Round 1

Reviewer 1 Report

The paper is mostly well written. The literature review still needs improvement. In addition to any other papers, if accessible, you may also consider to look for example:

1. Umer, W., Li, H., Szeto, G.P.Y. and Wong, A.Y.L., 2016. Identification of biomechanical risk factors for the development of lower-back disorders during manual rebar tying. Journal of Construction Engineering and Management143(1), p.04016080.

2. Umer, W., Li, H., Szeto, G.P.Y. and Wong, A.Y., 2017. Low-cost ergonomic intervention for mitigating physical and subjective discomfort during manual rebar tying. Journal of Construction Engineering and Management143(10), p.04017075.

3. Antwi-Afari, M.F., Li, H., Edwards, D.J., Pärn, E.A., Seo, J. and Wong, A., 2017. Effects of different weights and lifting postures on balance control following repetitive lifting tasks in construction workers. International Journal of Building Pathology and Adaptation35(3), pp.247-263.

4. Umer, W., Li, H., Szeto, G.P.Y. and Wong, A.Y., 2018. Proactive safety measures: quantifying the upright standing stability after sustained rebar tying postures. Journal of Construction Engineering and Management144(4), p.04018010.

5. Umer, W., Li, H., Lu, W., Szeto, G.P.Y. and Wong, A.Y., 2018. Development of a tool to monitor static balance of construction workers for proactive fall safety management. Automation in Construction94, pp.438-448

6. Yu, Y., Li, H., Umer, W., Dong, C., Yang, X., Skitmore, M. and Wong, A.Y., 2019. Automatic Biomechanical Workload Estimation for Construction Workers by Computer Vision and Smart Insoles. Journal of Computing in Civil Engineering33(3), p.04019010.

The conclusion can be part of the discussion as 'Discussion and Conclusion'.

Author Response

Thank you for all comments and suggestions from associated editor and reviewers.

The corrections have been done according to the peer reviewers’ and editorial team comments. The changes in the text of the manuscript and our answers to the comments from both reviewers are in red or using word corrector.

Please, see below the answer to reviewers:

Comments from the editors and reviewers:

-REVIEWER 1

 Overall Comments:

The paper is mostly well written. The literature review still needs improvement.

Thank you so much for your time and collaboration to review our manuscript.

Line 322. The conclusion can be part of the discussion as 'Discussion and Conclusion'.

ANSWER: Done.

Reviewer 2 Report

The presented paper tackles an interesitng subject, however, its results are easy to predict. Nevertheless, it is worth to study this problem deeper to prove its scientifically.

I would suggest to extend the limitation of the study by the fact that the investigated situation refers only to experimental investigation. Different results might be achieved in practical environment.

From practical point of view it might be interested to underline the problem of the quality of the product delivered in both barrels. It could have an influence on practical use of the recyclable barrels and lowering hazard to people working in such enviroment. 

It is needed to adjust the paper to the editorial journal requirements. For example, please check where should be located caption of the figures.

In Table 1 there is "a. Post Hoc analysis" - no need to indicate a. if there is no b.

Conclusions should refer to the results of the study. The present version is very general one.

Author Response

Thank you for all comments and suggestions from associated editor and reviewers.

The corrections have been done according to the peer reviewers’ and editorial team comments. The changes in the text of the manuscript and our answers to the comments from both reviewers are in red or using word corrector.

Please, see below the answer to reviewers:

Comments from the editors and reviewers:

-REVIEWER 2

 Overall Comments:

The presented paper tackles an interesitng subject, however, its results are easy to predict. Nevertheless, it is worth to study this problem deeper to prove its scientifically.

Thank you so much for your time and collaboration to review our manuscript.

Line 374.  Rewording suggested “to extend the limitation of the study by the fact that the investigated situation refers only to experimental investigation”

ANSWER: Done.

Rewording suggested about the problem “to underline the problem of the quality of the product delivered in both barrels. It could have an influence on practical use of the recyclable barrels and lowering hazard to people working in such enviroment.”

ANSWER: We not evaluated a rate of use or quality of each which, but only the biomechanics responses. This could be investigated in the future.

Line 130/203/236. to adjust the paper to the editorial journal requirements “caption of the figures.”

ANSWER: Done.

Line 291. “In Table 1 there is "a. Post Hoc analysis" - no need to indicate a. if there is no b.”

ANSWER: Done.

Line 381. Rewording suggested about the conclusion “The present version is very general one.”

ANSWER: Done

Reviewer 3 Report

Introduction. Please explain the significance of elevated and compressive forces (line 45).

Method. Standard deviation appears twice. Review the format (lines 69-72).

There are some English and grammar mistakes.

Author Response

Thank you for all comments and suggestions from associated editor and reviewers.

The corrections have been done according to the peer reviewers’ and editorial team comments. The changes in the text of the manuscript and our answers to the comments from both reviewers are in red or using word corrector.

Please, see below the answer to reviewers:

Comments from the editors and reviewers:

-REVIEWER 3

Thank you so much for your time and your collaboration to review our manuscript.

Line 45. “Introduction. Please explain the significance of elevated and compressive forces”

ANSWER: Done

Line 69-72. “Method. Standard deviation appears twice. Review the format”

ANSWER: Done

-

Reviewer 4 Report

 Thank you for the invitation to review this interesting manuscript. This study investigated the differences in effects of two lifting two types of barrels, recyclable versus steel, on neuromuscular and postural trunk responses in workers young workers. The study sample was composed of 12 active males who performed four tasks on a force platform while measuring their postural response with a force platform and their kinematic response by photography. The four tasks were lifting the two barrels full and lifting the two barrels empty. The authors showed that the steel barrel was impacting the kinetics and kinematics outcomes negatively. Overall enthusiasm for the manuscript is tempered by several issues mostly related the statistical design and the interpretation of some of the results. 

Major comments

1) The statistical design does not seem to be the most appropriate to test the hypothesis. A 2x2 design (type of barrel x barrel condition) would have been better to test the impact of lifting two types of barrels.

2) The kinematic and kinetic evaluations should be better explained, especially because the photographer could not be blinded to the condition. At what time point the photograph was taken? When did the platform start recording? As soon as the participant stepped on the platform? How kinematic data were normalized should also be better explained. The reason for normalization by total displacement is not clear. This was not used in the referenced paper (da Silva 2013, or Shigaki 2017 did not normalize) please explain the reason of such normalization with references to articles using the same normalization and how it can affect the data.

3) The interpretation of the neuromuscular findings should be better explained. In line 151 the authors stated that “All measurements were performed with reference to the 180 second reference of Smith and colleagues (Smith et al. 2011)” however Smith et al. do not seem to use such a reference. My concern is however on the interpretation of the results. As the authors stated that 180 degrees is the reference value for all measurements, would it more appropriate to analyses the results as “deviation from this reference”? With this in mind, it is not clear to me how “steel barrel produced greater vertical trunk flexion than the full recyclable barrel” while it was only deviated from 2 degrees from 180 degrees, while the full PET barrel deviated from the reference of 11 degrees. The results in table 3 seem to mean that the full PET barrel induced more deviation from reference while the opposite seems to be suggested in the text. 

Minor comments

4) According to the pedometer analysis (L165), most participants were “very active,” and not “active” as stated in the results section (L213). 

5) More precision on the maximum voluntary contraction tests used should be presented. Was maximal strength recorded? If so, the results should be displayed.  

6) The total displacement of the foot pressure center was recorded but not presented. This could be an outcome of interest. 

7)  The authors stated that “The validity and reliability (ICC > 0.80) of this parameter was confirmed in both elderly and healthy adults (da Silva et al. 2013)” (L135). However, Da Silva et al 2013 refers to a one leg balance test, and the ICC>80 was only found in older adults. This seems far from the experimental conditions of the present paper. 

8) The authors stated that “Significant differences were found among the 4 conditions for the three main variables analyzed” (L216); however table 1 do not show significant differences in the three variables. 

9) Table 1 presents an error in the labeling of variables. Moreover, the authors could add in the table that ES (d) is for full steel barrel vs. full PET barrel. Also, the reason for presenting Effect size of a not significant difference while table 2 do not present them is not clear. 

10) Borg scales are significant results, and the mean values for each condition could be presented. 

11) The limit section is small, especially when considering the different issues presented above. 

Author Response

Thank you for all comments and suggestions from associated editor and reviewers.

The corrections have been done according to the peer reviewers’ and editorial team comments. The changes in the text of the manuscript and our answers to the comments from both reviewers are in red or using word corrector.

Please, see below the answer to reviewers:

Comments from the editors and reviewers:

-REVIEWER 4

 Overall Comments:

Thank you for the invitation to review this interesting manuscript. This study investigated the differences in effects of two lifting two types of barrels, recyclable versus steel, on neuromuscular and postural trunk responses in workers young workers. The study sample was composed of 12 active males who performed four tasks on a force platform while measuring their postural response with a force platform and their kinematic response by photography. The four tasks were lifting the two barrels full and lifting the two barrels empty. The authors showed that the steel barrel was impacting the kinetics and kinematics outcomes negatively. Overall enthusiasm for the manuscript is tempered by several issues mostly related the statistical design and the interpretation of some of the results.

Thank you so much for your time and collaboration to review our manuscript.

Line ???.  The statistical design does not seem to be the most appropriate to test the hypothesis. A 2x2 design (type of barrel x barrel condition) would have been better to test the impact of lifting two types of barrels.

ANSWER: Thank you for this information, but as we used a control condition for each which, we assumed a single comparison between 4 conditions to determine the effects randomized and repeated measures of wearing external load over trunk on postural and neuromuscular responses. If we had an experimental group as individuals with CLBP, then a 2x2 design could be better employed. The power and sample size of our study is better for one-way ANOVA.

Line ???. The kinematic and kinetic evaluations should be better explained, especially because the photographer could not be blinded to the condition. At what time point the photograph was taken? When did the platform start recording? As soon as the participant stepped on the platform? How kinematic data were normalized should also be better explained. The reason for normalization by total displacement is not clear. This was not used in the referenced paper (da Silva 2013, or Shigaki 2017 did not normalize) please explain the reason of such normalization with references to articles using the same normalization and how it can affect the data.

ANSWER: Thank you for your question. In the previous studies, we not normalized the data because the object maintained in the trunk during postural control measures was the same across all experimental condition and with standard design measure, which is very different of barrels. To avoid bias or errors on measurement with our experience in different studies of our team in postural control, we assumed a new procedure of normalization from COP measurement to better report our results as in the EMG. Thus, we reformulated this section.

Please see references:

https://scholar.google.com/citations?user=9cYh4GoAAAAJ&hl=en

Line ???. The interpretation of the neuromuscular findings should be better explained. In line 151 the authors stated that “All measurements were performed with reference to the 180 second reference of Smith and colleagues (Smith et al. 2011)” however Smith et al. do not seem to use such a reference. My concern is however on the interpretation of the results. As the authors stated that 180 degrees is the reference value for all measurements, would it more appropriate to analyses the results as “deviation from this reference”? With this in mind, it is not clear to me how “steel barrel produced greater vertical trunk flexion than the full recyclable barrel” while it was only deviated from 2 degrees from 180 degrees, while the full PET barrel deviated from the reference of 11 degrees.The results in table 3 seem to mean that the full PET barrel induced more deviation from reference while the opposite seems to be suggested in the text.

ANSWER: Thank you for this information. We reformulated this section.

Line 165/213. According to the pedometer analysis (L165), most participants were “very active,” and not “active” as stated in the results section (L213).

ANSWER: Done.

Line???. More precision on the maximum voluntary contraction tests used should be presented. Was maximal strength recorded? If so, the results should be displayed. 

ANSWER: The manuscript is so long with so much information, thus we assumed to validate of this measures, but to avoid a description large of this test. This measure was only used for normalization of EMG. However, this information was detailed in the lines 231-236.

Line???. The total displacement of the foot pressure center was recorded but not presented. This could be an outcome of interest.

ANSWER:  We used only the main valid and reliable COP parameter and total displacement was used only on factor of correction. Thus we assumed the validity of data from area of COP and velocity.

Ref:

1.         Pinsault N, Vuillerme N. Test-retest reliability of centre of foot pressure measures to assess postural control during unperturbed stance. Medical engineering & physics. 2009;31(2):276-86.

Line 135. The authors stated that “The validity and reliability (ICC > 0.80) of this parameter was confirmed in both elderly and healthy adults (da Silva et al. 2013)” (L135). However, Da Silva et al 2013 refers to a one leg balance test, and the ICC>80 was only found in older adults. This seems far from the experimental conditions of the present paper.

ANSWER:  We changed this reference.

Line 216. The authors stated that “Significant differences were found among the 4 conditions for the three main variables analyzed” (L216); however table 1 do not show significant differences in the three variables.

ANSWER: We revised and included in the bold the p value.

Line ???. Table 1 presents an error in the labeling of variables. Moreover, the authors could add in the table that ES (d) is for full steel barrel vs. full PET barrel. Also, the reason for presenting Effect size of a not significant difference while table 2 do not present them is not clear.

ANSWER:  We revised.

Line???. Borg scales are significant results, and the mean values for each condition could be presented.

ANSWER: Not used this results.

Line???. The limit section is small, especially when considering the different issues presented above.

ANSWER: We revised.